# Learning Residual Set Representations for Mechanistic Gene Regulation

**Abstract**

Learning biologically meaningful representations requires alignment between model outputs and underlying regulatory mechanisms. In gene regulatory networks (GRNs), transcriptional regulation is inherently combinatorial: target genes are controlled by specific sets of interacting regulators. However, most machine learning approaches rely on edge-level representations that fail to capture this regulatory logic and are evaluated using pairwise objectives.

We formulate target-specific GRN inference as **Exact Regulator Set Recovery** and introduce a two-stage framework that separates candidate retrieval from set-level reasoning. Stage-1 retrieves a high-recall candidate pool using a target-conditioned attention retriever. Stage-2 selects the regulator team using **Residual HOS2**, which models non-additive interactions as a correction to decomposable pairwise evidence. Experiments on single-cell data with known combinatorial ground truth show that this residual set formulation enables more stable learning and improves recovery of mechanistically meaningful regulator teams in sparse genomic regimes.

## 1 Introduction

Gene regulatory networks (GRNs) govern cellular function through directed interactions between transcription factors and target genes. Single-cell transcriptomics provides unprecedented resolution for GRN inference, but also introduces sparsity, noise, and confounding effects that make causal discovery difficult [6,13]. Recent deep learning systems, including multiome-integrative approaches [23] and cell-specific GRN models [21], have improved *pairwise* edge prediction. Yet edge ranking is misaligned with a central biological reality: transcription factors rarely act alone. Gene regulation is mediated by interacting groups of transcription factors whose joint activity defines functional regulatory mechanisms [11,12,14,18], rather than by independent regulator–target effects [3,5,8,15,16]. As a result, representations that decompose regulation into independent edges fail to align with the mechanistic level at which biological function is defined.

From a representation learning perspective, this misalignment implies that high-performing models may still encode biologically uninformative abstractions, despite strong predictive accuracy. These results suggest that evaluation design, rather than model capacity alone, is a primary barrier to progress in genomic machine learning.

**Why edge ranking is not enough.** In combinatorial regulation, a model can produce a plausible ranked list and still miss the functional team: recovering most—but not all—true regulators yields an incomplete and potentially misleading mechanism. This failure mode is amplified by hub confounders, where genes with strong marginal correlation dominate pairwise scores even when the true regulatory logic depends on a specific interacting set.

**Proposed formulation: GRN inference as set recovery.** We reframe target-specific GRN inference as **Exact Regulator Set Recovery**. For each target gene $t$, we seek the exact regulator set $\mathcal{R}^\star(t)$ rather than a ranked list of edges. We propose a two-stage **Filter-and-Refine** pipeline:

1. **Stage-1 (Retrieval):** produce a high-recall candidate pool $\mathcal{P}_t$ (Top-$K$).

2. **Stage-2 (Selection):** score subsets $S \subseteq \mathcal{P}_t$ of size $R$ and output $\widehat{\mathcal{R}}(t)$.

**Key finding: a sparse-regime optimization barrier.** A natural approach is to apply a Set Transformer [9] to score candidate sets. We find that in this sparse genomic regime, "pure" set attention (without a decomposable anchor) can generalize poorly and remain far below residual set scoring, even after basic stabilization attempts (warmup / lower LR; Appendix E). These results motivate a residual formulation that (i) anchors optimization to robust additive evidence and (ii) lets attention focus exclusively on non-additive synergy.

**Contributions.**

- We formulate target-specific GRN inference as **Exact Regulator Set Recovery**, separating retrieval ceilings from set-level reasoning.
- We introduce a **context-aware attention retriever** that substantially improves candidate coverage.
- We propose **Residual HOS2**, a residual set scorer that stabilizes sparse training by learning non-additive interactions on top of a decomposable base.
- We provide controlled evidence on SERGIO DS3, including oracle injection, showing that residual high-order modeling is necessary to recover regulatory logic.

## 2 Related Work

**Pairwise GRN inference.** Most single-cell GRN benchmarks and algorithms evaluate pairwise edge recovery using edge-centric metrics, which quantify marginal associations but do not assess whether learned representations correspond to meaningful regulatory mechanisms [10, 13].

Modern deep-learning approaches improve edge inference using multiome integration and cell-specific modeling [21, 23], as well as representation learning and contrastive objectives [22]. However, these systems remain bound to pairwise metrics, which do not measure recovery of cooperative regulator teams [12, 18].

**High-order structure in genomics.** Hypergraph and higher-order representations have been explored for GRN inference and related tasks, motivated by the recognition that biological networks exhibit non-pairwise structure that cannot be faithfully represented by edge-level abstractions [1, 2, 17, 19, 20].

These methods often model global structure, whereas our goal is *target-specific* team discovery under a strict exact-match metric.

**Permutation-invariant set learning.** Deep Sets [24] and Set Transformers [9] provide principled architectures for learning on unordered collections. Our work highlights an under-emphasized regime: *sparse supervision + noisy features*, where a residual inductive bias can materially improve training stability and sample efficiency.

## 3 Problem Setup

Let $G$ denote the set of genes. For each target gene $t$, there is a ground-truth regulator set $\mathcal{R}^\star(t) \subset G$ of size $R = |\mathcal{R}^\star(t)|$. The goal is to predict a set $\widehat{\mathcal{R}}(t)$ of size $R$.

**Exact set recovery metric.** We evaluate **Unconditional Exact Match**:

$$\text{Acc} \;=\; \mathbb{E}_t\Big[\mathbb{I}\big(\widehat{\mathcal{R}}(t) = \mathcal{R}^\star(t)\big)\Big].$$

**Two-stage factorization and ceilings.** Enumerating all possible regulator teams is intractable ($O(|G|^R)$). We therefore decouple the problem into (i) high-recall retrieval of $\mathcal{P}_t$ and (ii) high-precision selection within $\mathcal{P}_t$. Unconditional accuracy is upper-bounded by **Recall@K**:

$$\text{Recall@}K \;=\; \mathbb{E}_t\Big[\mathbb{I}\big(\mathcal{R}^\star(t) \subseteq \text{Top-}K(t)\big)\Big].$$

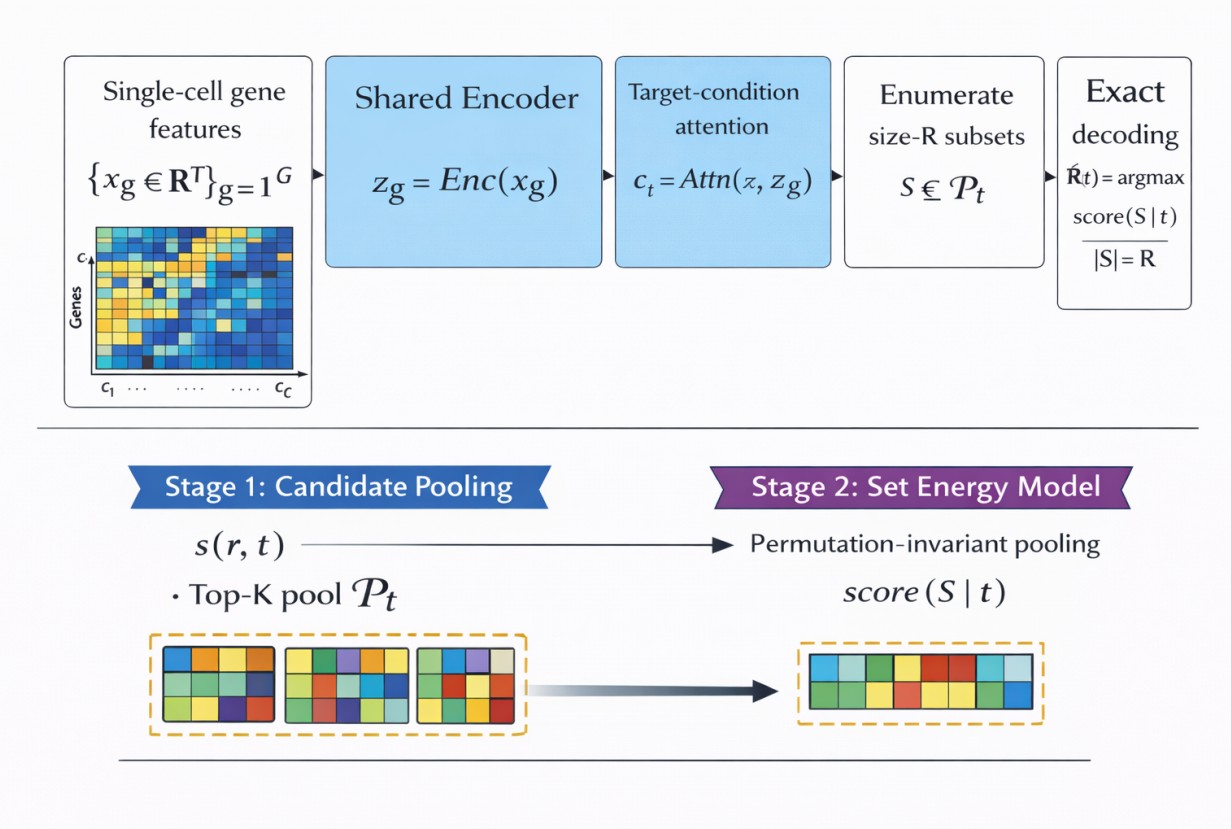

Figure 1: **Method overview: two-stage filter-and-refine pipeline for exact regulator set recovery.** Given gene expression features, Stage-1 retrieves a high-recall candidate pool of regulators for each target gene using a target-conditioned attention retriever. Stage-2 selects the regulator team by scoring candidate sets with a decomposable pairwise base and a residual high-order set module, followed by exact subset decoding. This stage-wise factorization separates retrieval ceilings from set-level reasoning and enables principled evaluation of combinatorial regulatory logic.

For exact subset decoding in Stage-2, we additionally cap the enumerated pool to $M_R$ for tractability, yielding an *effective* ceiling:

$$\text{Coverage@}M_R = \mathbb{E}_t\Big[\mathbb{I}\big(\mathcal{R}^\star(t) \subseteq \mathcal{P}_t[: M_R]\big)\Big].$$

We report Recall@80 (retrieval quality), Coverage@$M_R$ (ceiling under the decoding protocol), and conditional exact recovery on covered targets.

# 4 Method

## 4.1 Overview

Given a target $t$, Stage-1 produces a candidate pool $\mathcal{P}_t$. Stage-2 scores candidate sets $S \subseteq \mathcal{P}_t$ with $|S| = R$ and outputs the best set:

$$\widehat{\mathcal{R}}(t) = \arg \max_{S \subseteq \mathcal{P}_t,\ |S|=R} \text{Score}(S, t).$$

## 4.2 Stage-1: Retrieval

Context-aware attention pooler (Stage-1). We retrieve candidate regulators using a target-conditioned attention mechanism. Each gene is encoded into a normalized embedding via a shared encoder. For a

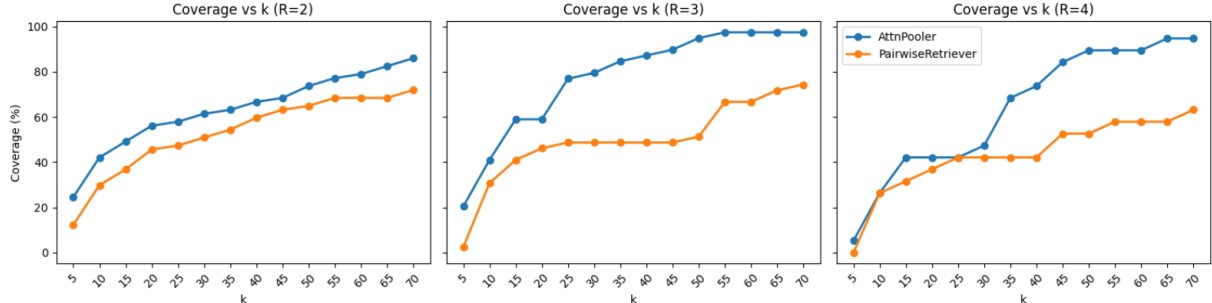

Figure 2: **Stage-1 retrieval coverage versus pool size.** Across regulator team sizes $R \in \{2, 3, 4\}$, the context-aware attention pooler (blue) consistently achieves higher coverage than a pairwise dot-product retriever (orange). The advantage increases with combinatorial complexity: for $R = 3$ and $R = 4$, the pairwise baseline exhibits an extended plateau at moderate coverage levels, while the attention pooler continues to improve and approaches near-saturation at moderate pool sizes. These results indicate that target-conditioned, contextual scoring is critical for reliably including all true regulators in the candidate pool, enabling accurate downstream exact set recovery.

given target gene, we compute a global context vector by attending from the target embedding to all gene embeddings, capturing dataset-level expression structure. Candidate regulators are then scored using an MLP over target, regulator, and context features, including interaction terms. Training uses a weighted multi-label objective to address sparsity. Full architectural and training details are provided in Appendix C.

## 4.3 Stage-2: Selection

Given $\mathcal{P}_t$, we compare decomposable (pairwise) selection, high-capacity decomposable selection, and residual set selection.

**Decomposable selection (PairS2).** Let $\phi_{\text{pair}}(r, t)$ denote a learned edge score (MLP on features formed from target/regulator embeddings and their interaction). A decomposable set score is:

$$\text{Score}_{\text{pair}}(S, t) \; = \; \sum_{r \in S} \phi_{\text{pair}}(r, t).$$

For a decomposable scorer, exact decoding reduces to selecting the Top-$R$ individual edges within the pool.

**Capacity ablation (Deep PairS2).** We match capacity to the set model by increasing depth in both the encoder and edge scorer while keeping the score decomposable:

$$\text{Score}_{\text{deep}}(S, t) \; = \; \sum_{r \in S} \phi_{\text{pair}}^{\text{deep}}(r, t).$$

This isolates whether depth/width alone resolves combinatorial logic.

**Residual High-Order Set Scorer (Residual HOS2).** We propose a residual decomposition:

$$\text{Score}(S, t) \; = \; \underbrace{\sum_{r \in S} \phi_{\text{pair}}(r, t)}_{\text{decomposable base}} \; + \; \underbrace{\psi_{\text{set}}(S, t)}_{\text{non-additive residual}} \; .$$

**Target-conditioned [CLS] and Set Transformer residual.** Let $z_t \in \mathbb{R}^d$ be the target embedding from the Stage-2 encoder. We define a target-conditioned [CLS] token as a learned affine map of $z_t$:

$$u_{\text{cls}}(t) = W_{\text{cls}} z_t + b_{\text{cls}} \in \mathbb{R}^d.$$

Given regulator embeddings $\{z_{r_i}\}_{i=1}^R$ for a candidate set $S = \{r_1, \ldots, r_R\}$, the residual module forms the sequence

$$U(S, t) = \big[u_{\mathrm{cls}}(t),\ z_{r_1}, \ldots, z_{r_R}\big],$$

applies $L$ layers of self-attention (Set Transformer encoder) [9], and maps the final [CLS] output to a scalar correction:

$$\psi_{\mathrm{set}}(S, t) = w^\top \mathrm{Enc}(U(S, t))_{\mathrm{cls}}.$$

**Zero-initialized correction head.** To stabilize learning, we initialize the final linear layer (the vector $w$ above) to all zeros, so $\psi_{\mathrm{set}}(S, t) = 0$ at initialization. Training thus begins as a well-behaved decomposable scorer and learns high-order structure as a correction.

**Set-ranking objective with curriculum negatives.** For Residual HOS2, we use a margin-ranking loss on sets: for each target $t$, we construct a true set $S^\star$ and a collection of negative sets $\{S_j^-\}$:

$$\mathcal{L}_{\mathrm{rank}}(t) \;=\; \frac{1}{J} \sum_{j=1}^J \mathrm{softplus}\Big(m - \big(\mathrm{Score}(S^\star, t) - \mathrm{Score}(S_j^-, t)\big)\Big),$$

with margin $m = 1.0$. Negatives follow a curriculum: early training mixes random negatives with near-miss negatives; later training emphasizes near-miss negatives formed by swapping one true regulator with a high-scoring confuser from the pool. This objective encourages the model to distinguish semantically meaningful regulator teams from superficially plausible but mechanistically incorrect alternatives under sparse supervision [7].

**Exact decoding under tractable pool sizes.** To remove inference as a confounder, we evaluate Residual HOS2 using **exact subset decoding** within a capped pool size $M_R$: $M_2 = 80$, $M_3 = 80$, $M_4 = 55$. We enumerate all $\binom{M_R}{R}$ subsets, score them in GPU batches, and return the argmax.

**Scalability note.** Exact decoding scales combinatorially in $R$ through $\binom{M_R}{R}$ and is therefore practical only for small $R$ under moderate caps (as in our $R \leq 4$ setting). Importantly, the *scoring* function itself scales polynomially in $R$ (a decomposable $O(R)$ base plus a Set Transformer residual over $R+1$ tokens), so larger regulator teams can be handled by replacing exhaustive enumeration with approximate decoding (e.g., beam search over incremental set construction, local swap-based refinement initialized from the decomposable base, or sample-and-rerank from a learned proposal). We leave a full evaluation of approximate decoding for larger $R$ to future work.

## 5 Experiments

### 5.1 Setup

We evaluate on **SERGIO DS3 (de-noised)** [4], a synthetic benchmark in which gene expression is generated from explicitly defined regulator sets, providing ground truth at the level of mechanistically meaningful regulatory units.

The dataset contains $G = 1200$ genes and $C = 2700$ cells (9 cell types $\times$ 300 cells/type). We use log1p-transformed expression features and z-score per gene (Appendix A). We stratify targets by $R \in \{2, 3, 4\}$ and use an 80/20 train/test split per stratum. Unless stated otherwise, results report mean±std over 3 splits/seeds (42/43/44).

### 5.2 Stage-1 Retrieval Quality: Recall@80

Table 1 reports **Recall@80** on the test set. The context-aware attention pooler substantially improves retrieval quality over a dot-product pairwise retriever.

Table 1: **Stage-1 retrieval quality (Recall@80)** on SERGIO DS3 test targets. Recall@80 measures whether the *entire* true regulator set is contained in the Top-80 candidate regulators for each target.

| R | Pairwise Dot-Prod Retriever | Context-Attn Pooler (Ours) | Δ |
|---|---|---|---|
| 2 | 0.6491 | 0.8947 | +0.2456 |
| 3 | 0.7179 | 0.8718 | +0.1538 |
| 4 | 0.8421 | 1.0000 | +0.1579 |

Table 2: **Exact Regulator Set Recovery on SERGIO DS3 (de-noised).** Stage-1 produces candidate pools of size $K=80$ using a **Context-Attn Retriever**; Stage-2 performs selection with an exact decoding cap $M_R = \{80, 80, 55\}$ for $R = \{2, 3, 4\}$. We report **Coverage@$M_R$** (fraction of test targets whose capped pool contains the full true set), **Unconditional Exact** (exact set match over all test targets), and **Conditional Exact** (exact set match restricted to covered targets). Values are mean±std over 3 seeds (42/43/44).

| Model | Metric | R=2 | R=3 | R=4 |
|---|---|---|---|---|
| **Coverage@$M_R$ (Stage-1 Context-Attn pools)** | | $0.895 \pm 0.018$ | $0.949 \pm 0.068$ | $0.930 \pm 0.061$ |
| Stage1TopR | Unconditional Exact | $0.058 \pm 0.027$ | $0.034 \pm 0.059$ | $0.018 \pm 0.030$ |
| | Conditional Exact | $0.065 \pm 0.029$ | $0.034 \pm 0.059$ | $0.020 \pm 0.034$ |
| PairS2 (decomposable) | Unconditional Exact | $0.251 \pm 0.056$ | $0.222 \pm 0.015$ | $0.140 \pm 0.061$ |
| | Conditional Exact | $0.281 \pm 0.064$ | $0.236 \pm 0.030$ | $0.153 \pm 0.072$ |
| **Residual HOS2 (ours)** | **Unconditional Exact** | $\mathbf{0.281 \pm 0.061}$ | $\mathbf{0.342 \pm 0.059}$ | $\mathbf{0.193 \pm 0.110}$ |
| | **Conditional Exact** | $\mathbf{0.314 \pm 0.068}$ | $\mathbf{0.360 \pm 0.048}$ | $\mathbf{0.209 \pm 0.126}$ |

## 5.3 Baselines and Ablations

We compare against:

1. **Stage1TopR:** pick Top-$R$ regulators from the Stage-1 pool (no set modeling).

2. **PairS2 (Shallow):** decomposable pairwise Stage-2 scorer.

3. **Deep PairS2:** high-capacity decomposable scorer (Seed 42).

4. **Residual HOS2 (Ours):** decomposable base + residual set attention trained with margin ranking.

5. **Appendix ablation (PureHOS2):** a non-residual Set Transformer scorer; see Appendix E.

## 5.4 Main Results: Exact Set Recovery

Table 2 reports **Exact Regulator Set Recovery**. **Coverage@$M_R$** depends only on the Stage-1 pool and the decoding cap $M_R$; it is identical for all Stage-2 models under the same Stage-1 retriever and caps. We use exact decoding for Residual HOS2 with $M_R = \{80, 80, 55\}$. For decomposable models, the optimal decoding is Top-$R$ edges within the same capped pool. Appendix G isolates the effect of Stage-1 retrieval by training the same PairS2 selector on pools produced by the pairwise dot-product retriever.

**Capacity is not the bottleneck.** Deep PairS2 does not resolve combinatorial logic recovery and remains low on $R = 4$, indicating that depth alone does not overcome additive ambiguity.

**Residual high-order modeling improves exact team recovery under ambiguity.** At $K = 80$, Residual HOS2 improves unconditional exact recovery over PairS2 for all complexities, with the strongest gains at higher order: $R = 3$: $0.342 \pm 0.059$ vs. $0.222 \pm 0.015$ and $R = 4$: $0.193 \pm 0.110$ vs. $0.140 \pm 0.061$. Importantly, conditional exact gains persist when restricting to covered targets, indicating improvements are not explained solely by retrieval ceiling effects.

Table 3: **Oracle Injection (Perfect Retrieval).** Exact set recovery when $\mathcal{R}^\star(t) \subseteq \mathcal{P}_t$ is guaranteed by injection and true-safe trimming. Decomposable models (PairS2) decode by Top-$R$ edges; Residual HOS2 performs exact search over subsets of size $R$ in $\mathcal{P}_t$.

| Model | R = 2 | R = 3 | R = 4 |
|---|---|---|---|
| Shallow PairS2 | 0.2456 | 0.1538 | 0.0526 |
| Deep PairS2 | 0.1579 | 0.1282 | 0.1053 |
| **Residual HOS2** | **0.3509** | **0.4359** | **0.3684** |

**Non-residual set scoring remains far below residual set scoring.** PureHOS2 can pass a memorization sanity check (near-zero training loss on a small target subset), confirming correct wiring, but generalizes poorly on unconditional exact recovery. A small stabilization sweep (warmup / lower LR) does not close the gap (Appendix E).

**Warm-start for PureHOS2.** PureHOS2 (Warm-start) copies *only* the **PairS2 encoder** weights into the PureHOS2 encoder (same MLP shape), then freezes/unfreezes; it does *not* inherit the Residual HOS2 residual module or scoring decomposition.

## 5.5   Oracle Injection: isolating set logic from retrieval and exposing hub confusers

A key difficulty in GRN inference is that *high pairwise association is not equivalent to correct regulatory logic*. In realistic regimes, many targets exhibit strong correlations with a small number of broadly co-expressed genes (*"hubs"* or *transitive confusers*). Pairwise edge predictors therefore tend to rank the same few genes highly across many targets, yielding plausible edge lists that nonetheless fail to recover the *specific non-additive combination* of regulators that functionally controls a given target.

**Oracle protocol.** To disentangle failures of *retrieval* from failures of *set selection*, we run an *oracle injection* evaluation. For each target $t$ with true regulator set $\mathcal{R}^\star(t)$ of size $R$, we construct a candidate pool $\mathcal{P}_t$ by taking the Stage-1 Top-$M$ candidates and *injecting* any missing true regulators, followed by *true-safe trimming* that never removes members of $\mathcal{R}^\star(t)$ (Appendix H). This guarantees $\mathcal{R}^\star(t) \subseteq \mathcal{P}_t$ for every evaluated target, so any remaining error must come from the Stage-2 model's inability to identify the correct set *even when the answer is present*.

**Why pairwise decoding fails under oracle retrieval.** For a strictly decomposable model (PairS2), the optimal set under additive scoring is always obtained by selecting the Top-$R$ edges individually:

$$\hat{S}_{\text{decomp}}(t) \; = \; \arg\max_{|S|=R} \sum_{r \in S} s_{\text{pair}}(r, t) \; = \; \text{Top-}R\big(\{s_{\text{pair}}(r, t)\}_{r \in \mathcal{P}_t}\big).$$

This decoding is fundamentally vulnerable to hub confusers: a broadly co-expressed confuser can crowd out weaker marginal regulators. Increasing MLP capacity does not remove this limitation, because the *additive* structure cannot represent cooperative effects in which a regulator becomes informative only in the presence of others.

**Oracle results: residual high-order modeling is necessary.** Table 3 reports exact set recovery under oracle injection on de-noised SERGIO DS3. The Residual HOS2 substantially outperforms both shallow and deep decomposable baselines, with gains growing sharply with regulatory complexity. In particular, for $R = 4$ the residual set model attains 0.3684 exact recovery vs. 0.1053 for the strongest decomposable baseline (Deep PairS2), a $\approx 3.5\times$ gap under perfect retrieval that directly isolates the value of modeling non-additive interactions.

**Real-data retrieval evaluation.** We additionally evaluate the Stage-1 retriever on a real mESC ChIP-seq gene regulatory network paired with single-cell expression data to assess scalability beyond synthetic benchmarks. Because real-world GRNs lack ground-truth regulator teams, this evaluation focuses on retrieval quality rather than exact set recovery. The context-aware attention retriever substantially improves strict regulator-set coverage over a pairwise baseline. Full experimental details and results are provided in Appendix I
.

# 6 Discussion and Limitations

**Residual sets as meaningful representations.** Learning meaningful biological representations requires alignment between model outputs and underlying mechanism. Pure set attention must rediscover low-level correlations before learning interaction structure, which can be unstable under sparse supervision. Residual learning stabilizes this process by explicitly separating additive evidence from interaction logic, yielding representations that more directly correspond to regulatory mechanisms.

**When interactions define the concept.** In some targets, marginal effects are sufficient to identify regulators, and decomposable scoring can perform well. However, the oracle injection analysis shows that when regulatory function depends on specific combinations, additive representations fail even under ideal retrieval. This highlights that regulatory logic is inherently a *set-level concept*, not an edge-level one.

**Evaluation and semantic alignment.** SERGIO enables exact semantic evaluation by providing ground-truth regulator sets that correspond directly to the mechanistic units of regulation, allowing representation quality to be assessed in terms of biological meaning rather than edge-level accuracy [4].

Real-world datasets typically provide edge-centric summaries that obscure context-specific regulatory teams [13]. Future work should explore evaluation through downstream functional alignment, such as perturbation response or expression prediction.

**Scalability of the representation.** Exact decoding is feasible here due to small regulator sets. For larger teams, approximate decoding strategies can be employed while preserving the learned residual set representation, suggesting a path toward scalable mechanistic modeling.

# 7 Conclusion

Meaningful representations of biological systems must reflect the level at which functional mechanisms operate. In gene regulation, this level is inherently set-valued: regulatory function emerges from interacting groups of factors rather than independent edges. We show that edge-level and purely additive representations fail to capture this structure under sparse supervision, while pure set attention is difficult to optimize. Residual HOS2 provides a principled compromise by learning regulatory teams as residual corrections to decomposable evidence, yielding representations aligned with mechanistic meaning. This perspective suggests that modeling interacting subsets may be a general strategy for learning semantically meaningful representations in complex biological systems. More broadly, this work suggests that learning meaningful representations of biological systems requires choosing representational units that align with the mechanisms by which biological function is realized.

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

# A    Feature Construction and Preprocessing

We use SERGIO DS3 de-noised expression with $G = 1200$ genes and at least $C = 2700$ cells. We construct RAW2700 features by applying $\log(1 + x)$ to expression and then z-scoring per gene across the first 2700 cells, yielding a feature vector in $\mathbb{R}^{2700}$ per gene.

# B    Ground-Truth GRN Construction

A key requirement of our evaluation is access to *exact* ground-truth regulator sets for each target gene. This appendix clarifies how ground-truth gene regulatory networks (GRNs) are defined in SERGIO and why they enable principled evaluation of combinatorial regulator recovery.

**Origin of Ground-Truth Regulatory Structure:**    SERGIO is a mechanistic gene expression simulator that generates single-cell RNA sequencing data from a predefined gene regulatory network [4]. The GRN is specified *a priori* as a directed graph in which each target gene is regulated by a fixed set of transcription factors. These regulator sets are provided explicitly as part of the dataset specification and are not inferred from expression data.

For each target gene $t$, SERGIO defines a regulator set

$$\mathcal{R}^{\star}(t) = \{r_1, \dots, r_R\},$$

where the regulator set size $R$ is fixed and known. Gene expression trajectories are then generated by simulating transcriptional dynamics governed by these regulatory relationships, followed by the injection of realistic single-cell noise processes.

**Combinatorial Regulation:**    Unlike pairwise GRN benchmarks that focus on independent regulator–target edges, SERGIO explicitly supports *combinatorial regulation*. Each target gene may depend on multiple regulators simultaneously, and its expression is governed by a nonlinear function of the joint regulator activity. As a result, the correct prediction target is not a ranked list of edges, but the *exact regulator set* associated with each gene.

In this work, we focus on targets with regulator set sizes $R \in \{2, 3, 4\}$, which correspond to increasing levels of combinatorial ambiguity. This setting is particularly challenging for additive or decomposable models, as many incorrect regulator combinations can explain marginal expression patterns equally well.

**Ground-Truth Labels and Evaluation Targets:**    Ground-truth regulator sets are extracted directly from the SERGIO interaction files, which list for each target gene the identities of all regulating transcription factors. These sets are treated as immutable ground truth throughout training and evaluation.

Importantly, while gene expression data are simulated, the regulator labels are not noisy or approximate. This allows us to evaluate exact regulator set recovery without confounding effects from label uncertainty or incomplete annotations.

**Implications for Evaluation:**    Because the ground-truth GRN is known exactly, SERGIO enables controlled evaluation of: (i) retrieval quality, measured by whether the true regulator set is included in a candidate pool, and (ii) reasoning quality, measured by exact recovery of the full regulator set.

This separation allows us to disentangle errors arising from candidate retrieval from those arising from higher-order set reasoning. In particular, conditional exact recovery isolates the ability of a model to resolve combinatorial ambiguity given sufficient candidate coverage.

While SERGIO is a simulated benchmark, it provides a necessary testbed for evaluating exact regulator team recovery—an objective that is currently infeasible to assess at scale using real scRNA-seq data due to incomplete ground-truth annotations.

# C   Stage-1 Architectures and Training Details

**Dot-product retriever:**   A shared MLP encoder maps gene features to $h_g$, then two linear heads map to $z_g^{(s)}$ and $z_g^{(t)}$. Training uses weighted BCE on multi-label regulator prediction for each target, with self-edges masked.

**Context-aware attention pooler:**   We encode genes to normalized embeddings $z_g$ and compute a target-conditioned context vector using multi-head attention with query $z_t$ over keys/values $\{z_g\}_{g \in G}$. Each candidate regulator $r$ is scored using a learned MLP on features that include $z_r$, $z_t$, $c_t$, interaction terms (elementwise products), and dot products (e.g., $\langle z_r, z_t \rangle$, $\langle z_r, c_t \rangle$). We train with weighted BCE, add small Gaussian noise to inputs during training for stability, and evaluate Recall@80.

Stage-1 maps each gene $g$ to an embedding and ranks candidate regulators for each target. We compare two retrievers.

**(Baseline) Dot-product pairwise retriever:**   We train a lightweight dual-head embedding model: a shared encoder produces a hidden representation $h_g$, then two projection heads produce a *source* embedding $z_g^{(s)}$ (regulator role) and a *target* embedding $z_g^{(t)}$ (target role). Candidates are scored by a dot product:

$$s(r \to t) \;=\; \langle z_t^{(t)}, z_r^{(s)} \rangle, \qquad \mathcal{P}_t \;=\; \text{Top-}K\{s(r \to t)\}.$$

Training uses a multi-label (BCE) objective over regulators for each target, with self-edges masked.

**Context-aware attention pooler (Stage-1):**   Let $x_g \in \mathbb{R}^D$ denote the feature vector of gene $g \in \{1, \ldots, G\}$. The model first produces a normalized embedding for every gene via a shared encoder

$$h_g = f_\theta(x_g) \in \mathbb{R}^d, \tag{1}$$

$$z_g = \frac{h_g}{\|h_g\|_2} \in \mathbb{R}^d. \tag{2}$$

For each target $t$, we compute a *target-conditioned global context* vector by attending from the target embedding $z_t$ (as the query) to the set of all gene embeddings $\{z_g\}_{g=1}^G$ (as keys/values). Using multi-head attention with $H$ heads, for head $h \in \{1, \ldots, H\}$ define

$$q_t^{(h)} = W_Q^{(h)} z_t, \qquad k_g^{(h)} = W_K^{(h)} z_g, \qquad v_g^{(h)} = W_V^{(h)} z_g, \tag{3}$$

$$\alpha_{t,g}^{(h)} = \frac{\exp\left(\langle q_t^{(h)}, k_g^{(h)} \rangle / \sqrt{d_h}\right)}{\sum_{j=1}^G \exp\left(\langle q_t^{(h)}, k_j^{(h)} \rangle / \sqrt{d_h}\right)}, \tag{4}$$

$$c_t^{(h)} = \sum_{g=1}^G \alpha_{t,g}^{(h)} v_g^{(h)}, \tag{5}$$

and concatenate heads to obtain the context

$$c_t = \text{Concat}\left(c_t^{(1)}, \ldots, c_t^{(H)}\right) \in \mathbb{R}^d. \tag{6}$$

(Equivalently, $c_t = \text{MHA}(z_t, Z, Z)$ where $Z = [z_1; \ldots; z_G]$.)

**Candidate scoring with target context:**   For each candidate regulator $r$, we construct an interaction feature map

$$\phi(z_r, z_t, c_t) = \left[z_r;\; z_t;\; c_t;\; z_r \odot z_t;\; z_r \odot c_t;\; z_t \odot c_t;\; \langle z_r, z_t \rangle;\; \langle z_r, c_t \rangle\right] \in \mathbb{R}^{6d+2}, \tag{7}$$

where $\odot$ denotes elementwise product. The Stage-1 logit for the directed pair $(r \to t)$ is produced by an MLP scorer $s_\psi(\cdot)$:

$$\ell_{t,r} = s_\psi(\phi(z_r, z_t, c_t)), \quad \ell_{t,t} = -\infty \;\; (\text{mask self-loop}). \tag{8}$$

Table 4: **PureHOS2 cheap stabilization sweep** (Seed 42). Despite passing a memorization sanity check, PureHOS2 remains far below Residual HOS2 on unconditional exact recovery, especially for $R \geq 3$.

| Setting (Seed 42) | R=2 | R=3 | R=4 |
|---|---|---|---|
| A: baseline (lr $2 \times 10^{-4}$, no warmup) | 0.0877 | 0.0000 | 0.0000 |
| B: warmup500 (lr $2 \times 10^{-4}$) | 0.0175 | 0.0513 | 0.0000 |
| C: lowerLR (lr $10^{-4}$, no warmup) | 0.0526 | 0.0256 | 0.0526 |

Finally, the candidate pool for target $t$ is the Top-$K$ regulators under these logits:

$$\mathcal{P}_t = \text{Top-}K\left(\{\ell_{t,r}\}_{r=1}^{G}\right). \tag{9}$$

**Training objective:** Let $y_{t,r} = \mathbb{I}[r \in \mathcal{R}^\star(t)]$ be the multi-label indicator of true regulators for target $t$. We train Stage-1 using a weighted binary cross-entropy objective over all regulators:

$$\mathcal{L}_{\text{S1}}(\theta, \psi) = -\sum_{t \in \mathcal{T}_{\text{train}}} \sum_{r=1}^{G} \left[w\, y_{t,r}\, \log \sigma(\ell_{t,r}) + (1 - y_{t,r}) \log\left(1 - \sigma(\ell_{t,r})\right)\right]. \tag{10}$$

where $\sigma$ is the sigmoid and $w > 0$ is a positive-class weight to address sparsity.

# D    Optimization Details and Hyperparameters

Stage-1 attention pooler: 20 epochs, AdamW, lr $2 \times 10^{-4}$, wd $10^{-3}$, batch of 16 targets, embedding dim 128, 4 attention heads, dropout 0.2, noise std 0.01. Stage-2: 15 epochs, AdamW, lr $2 \times 10^{-4}$, wd $10^{-3}$, batch of 16 targets. Residual HOS2 uses margin $m = 1.0$, negative curriculum (random $\rightarrow$ near-miss), and exact decoding with caps $M_2 = 80$, $M_3 = 80$, $M_4 = 55$.

# E    PureHOS2 Stabilization Ablation

We evaluate a non-residual set scorer (PureHOS2) that directly scores sets with a Set Transformer (target-conditioned [CLS] token) and the same margin-ranking objective as Residual HOS2, but *without* a decomposable base term. A memorization sanity check reaches near-zero loss, confirming correct wiring; however, generalization remains poor. We perform a small stabilization sweep (Seed 42): baseline, LR warmup (500 steps), and lower LR.

# F    Capacity Ablation: Deep PairS2 (Single Seed)

A natural hypothesis is that the gap between decomposable scoring and Residual HOS2 could be closed by simply increasing model capacity (depth/width) while keeping the score decomposable. We therefore evaluate a **Deep PairS2** variant that increases encoder/scorer depth (implementation-matched to the set model's capacity budget) but retains an additive set score:

$$\text{Score}_{\text{deep}}(S, t) = \sum_{r \in S} \phi_{\text{pair}}^{\text{deep}}(r, t).$$

Because decoding for decomposable models is optimally Top-$R$ edge selection within the capped pool, this ablation isolates *capacity* from *inductive bias*.

Table 6: **PairS2 under different Stage-1 candidate pools** on SERGIO DS3 (de-noised). Coverage@$M_R$ depends only on Stage-1 retrieval and the cap $M_R$, while unconditional/conditional exact quantify Stage-2 selection performance. PairS2 is identical in both blocks; only the Stage-1 retriever (pool generator) changes.

| Stage-1 Pools | Metric | R=2 | R=3 | R=4 |
|---|---|---|---|---|
| **Pairwise Dot-Prod Pools** | Coverage@$M_R$ | $0.731 \pm 0.037$ | $0.752 \pm 0.015$ | $0.649 \pm 0.080$ |
| | Unconditional Exact | $0.222 \pm 0.010$ | $0.274 \pm 0.065$ | $0.053 \pm 0.053$ |
| | Conditional Exact | $0.305 \pm 0.028$ | $0.363 \pm 0.080$ | $0.079 \pm 0.084$ |
| **Context-Attn Pools** | Coverage@$M_R$ | $0.895 \pm 0.018$ | $0.949 \pm 0.068$ | $0.930 \pm 0.061$ |
| | Unconditional Exact | $0.251 \pm 0.056$ | $0.222 \pm 0.015$ | $0.140 \pm 0.061$ |
| | Conditional Exact | $0.281 \pm 0.064$ | $0.236 \pm 0.030$ | $0.153 \pm 0.072$ |

Table 5: **Deep PairS2 capacity ablation** on SERGIO DS3 (de-noised), **Seed 42**. We report **Unconditional Exact** set recovery under the same Stage-2 decoding protocol. Despite substantially increased depth, Deep PairS2 does not close the gap to Residual HOS2 on complex targets, indicating that the remaining errors stem from the limitations of decomposable scoring rather than insufficient capacity.

| Model (Seed 42) | R=2 | R=3 | R=4 |
|---|---|---|---|
| Deep PairS2 (deeper decomposable) | 0.123 | 0.179 | 0.053 |

# G  Stage-1 Pooler Ablation: PairS2 on Pairwise vs Context-Attn Pools

To isolate the effect of **Stage-1 retrieval** from **Stage-2 selection**, we train the *same* decomposable selector (PairS2 SAFE) on candidate pools produced by two different retrievers: (i) the **pairwise dot-product retriever** and (ii) the **context-aware attention pooler**. All settings (features, caps $M_R = \{80, 80, 55\}$, negatives, epochs) are held fixed. We report **Coverage@$M_R$**, **Unconditional Exact**, and **Conditional Exact** on the test set (mean±std over 3 seeds: 42/43/44).

**Interpretation.** Switching from pairwise dot-product pools to context-aware attention pools substantially increases Coverage@$M_R$ (the retrieval ceiling), especially for larger $R$. This confirms that improved candidate retrieval is necessary to make exact set recovery feasible under capped decoding.

# H  Oracle injection protocol and decomposable Top-$R$ optimality

**Oracle pool construction:**  For each target $t$, we start from the Stage-1 Top-$M$ list, inject any missing members of $\mathcal{R}^\star(t)$, and apply true-safe trimming that never removes regulators in $\mathcal{R}^\star(t)$, yielding $\mathcal{P}_t$ with $\mathcal{R}^\star(t) \subseteq \mathcal{P}_t$ by construction.

**Why Top-$R$ decoding is optimal for decomposable scores:**  If $\text{Score}(S, t) = \sum_{r \in S} s(r, t)$, then the maximizing size-$R$ subset is obtained by taking the $R$ largest individual scores:

$$\arg \max_{|S|=R} \sum_{r \in S} s(r, t) = \text{Top-}R(\{s(r, t)\}_{r \in \mathcal{P}_t}).$$

# I  Real-World Retrieval Evaluation on mESC

We additionally evaluate Stage-1 retrieval on a real gene regulatory network derived from an mESC ChIP-seq network paired with a single-cell expression matrix. The expression matrix contains $G$=18,386 genes and $C$=421 cells, and the candidate regulator set contains $|\mathcal{R}|$=229 regulators. After gene-name mapping and

Table 7: **mESC Stage-1 retrieval (regs-only) on 1,500 test targets.** StrictCov@K requires the *full* regulator set to be included in Top-$K$; EdgeRec@K is the average fraction of true regulators recovered within Top-$K$.

| | Pairwise | | Context-Attn (Ours) | |
|---|---|---|---|---|
| **K** | **StrictCov@K** | **EdgeRec@K** | **StrictCov@K** | **EdgeRec@K** |
| 100 | 0.0627 | 0.8960 | 0.0727 | 0.9050 |
| 120 | 0.1227 | 0.9385 | 0.1627 | 0.9510 |
| 140 | 0.2633 | 0.9665 | 0.3980 | 0.9791 |
| 160 | 0.4247 | 0.9780 | 0.7040 | 0.9933 |
| 180 | 0.5567 | 0.9836 | 0.9047 | 0.9979 |
| 200 | 0.6873 | 0.9900 | 0.9893 | 0.9998 |

filtering, the directed network contains 806,646 kept edges and 12,284 target genes (mean/median/max in-degree: 65.7/71/182). We randomly split targets into 4,000 train targets and 1,500 test targets.

**Metrics (regs-only ranking):** Because the regulator universe is known ($|\mathcal{R}|$=229), we rank *only* regulators for each target and report two complementary retrieval metrics: (i) **Strict Coverage@K**, the fraction of targets whose *entire* ChIP regulator set is contained in the Top-$K$ ranked regulators,

$$\text{StrictCov@}K = \mathbb{E}_t \left[ \mathbb{I}\big( \mathcal{R}^\star(t) \subseteq \text{Top-}K(t) \big) \right],$$

and (ii) **Edge-Recall@K**, the per-target fraction of true regulators recovered within the Top-$K$ list, averaged over targets,

$$\text{EdgeRec@}K = \mathbb{E}_t \left[ \frac{|\mathcal{R}^\star(t) \cap \text{Top-}K(t)|}{|\mathcal{R}^\star(t)|} \right].$$

We compare a dot-product **pairwise retriever** against our **context-aware attention pooler**.

**Results:** Figure 3 shows retrieval performance as $K$ increases from 100 to 200. While both methods achieve high edge-level recall at moderate $K$, the attention pooler dramatically improves *strict* set coverage for these large regulator sets. At $K$=200, the attention pooler reaches near-saturation strict coverage (0.989) whereas the pairwise retriever remains substantially lower (0.687), indicating that contextual scoring is critical for reliably including the full long-tail regulator set in dense real networks.

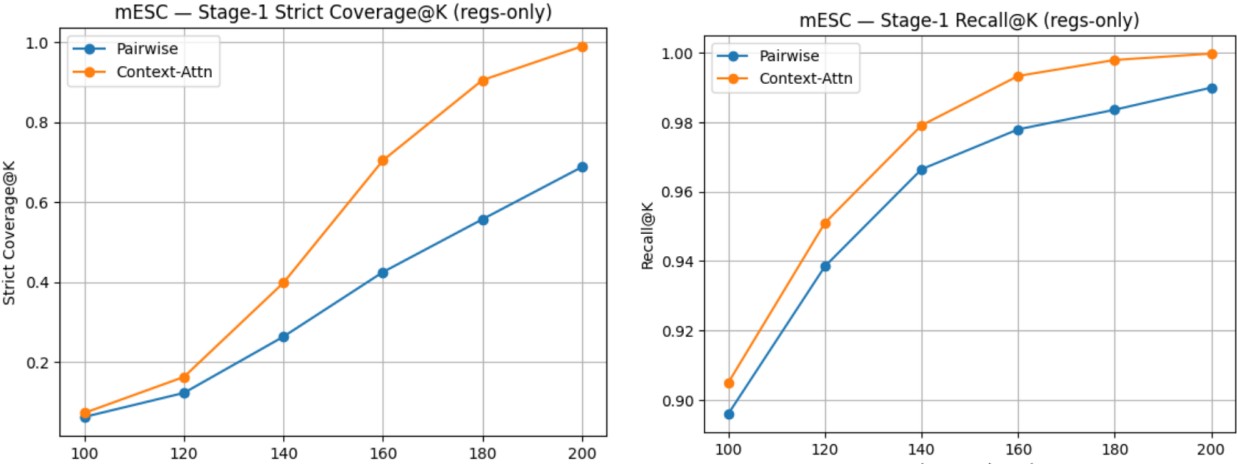

Figure 3: **mESC Stage-1 retrieval scaling (regs-only).** Left: Strict Coverage@K (full regulator set contained in Top-$K$). Right: Edge-Recall@K (fraction of true regulators in Top-$K$). The context-aware attention retriever substantially improves strict coverage for large regulator sets, approaching saturation by $K$=200.

**Scope of this real-data evaluation:** Unlike SERGIO (where $R \leq 4$ enables exact subset decoding and strict exact-match evaluation), mESC targets often have large regulator sets ($R \approx 70$ median), making exhaustive Stage-2 decoding infeasible. We therefore use mESC primarily to validate the *retrieval* component and highlight how contextual scoring improves the candidate ceiling in dense real GRNs. A full large-$R$ Stage-2 evaluation requires approximate decoding and additional biological validation, which we leave to future work.

