# OpenReview forum: "Learning Residual Set Representations for Mechanistic Gene Regulation"
_ICLR.cc/2026/Workshop/LMRL — Submitted to ICLR 2026 Workshop LMRL_

### Official Review · Reviewer_ZJYW · 2026-02-22
**Review of "Learning Residual Set Representations for Mechanistic Gene Regulation"**

**Rating:** 6
**Confidence:** 4

**Review:**

Summary:
- This paper proposes to formulate target-specific GRN inference as an Exact Regulator Set Recovery problem, arguing that conventional pairwise edge-ranking approaches are fundamentally misaligned with the underlying combinatorial nature of transcriptional regulation. The authors propose a two-stage filter-and-refine approach: (1) a context-aware attention pooler that uses target-conditioned attention for top-K candidate retrieval, and (2) Residual HOS2, which scores candidate sets using additive pairwise scores plus a non-additive correction learned with a Set Transformer. Experiments are primarily conducted on synthetic data to demonstrate benefits over the authors' baselines. Additionally, oracle injection experiments show how the proposed method better handles challenging cases with highly co-expressed confounding genes.

Strengths:
- This work attempts to address an important limitation of focusing on edge rankings. Conceptually, set-level recovery indeed better captures the underlying biology, with regulators frequently acting combinatorially.
- The oracle injection experiment is quite interesting, further demonstrating the limitations of decomposable models and motivating the approach using examples with highly co-expressed confounders.

Weaknesses:
- Evaluation is mostly limited to synthetic data. While synthetic benchmarks can be useful for demonstrating key concepts, it is important to validate on real-world data despite the associated challenges. Results on synthetic data may not be representative of performance in realistic settings.
- The paper only compares against the authors' own baselines, but I think it is crucial to also compare against established methods. Even though many existing approaches were not designed for exact set recovery, it should be possible to adapt them using simple set selection strategies. This would have better contextualized the shortcomings of existing methods and the significance of the proposed Residual HOS2.
- The overall writing could be improved. While the paper appears polished, I found many sentences to be vague and difficult to follow. Prioritizing clarity over abstract language would help readers more easily understand the key motivations and core contributions.

---

### Official Review · Reviewer_aerE · 2026-02-23

**Rating:** 6
**Confidence:** 3

**Review:**

This paper reformulates gene regulatory network (GRN) inference as Exact Regulator Set Recovery, arguing that mechanistic gene regulation is inherently combinatorial and not well captured by edge-level objectives. It proposes a two-stage filter-and-refine framework with a residual high-order set scorer (Residual HOS2) that models non-additive interactions as a correction to decomposable pairwise evidence, providing a compelling inductive bias for sparse supervision. Experiments on the synthetic SERGIO DS3 benchmark show consistent improvements over decomposable baselines, particularly under oracle retrieval, and the ablations carefully disentangle retrieval ceilings, capacity, and inductive bias. However, evaluation of exact set recovery is limited to small regulator sets (R ≤ 4) in simulation, without much results on real-data experiments, leaving questions about scalability and practical impact in realistic GRNs.

---

### Official Review · Reviewer_qjhM · 2026-02-24
**The ideas are well motivated, but paper lacks comparison to established baselines**

**Rating:** 5
**Confidence:** 5

**Review:**

The authors propose Residual Set Representation Learning for recovering the regulatory sets in a GRN as opposed to recovering individual edges. The design of the model is well justified throughout the paper, and authors include benchmarks on both simulated and real single-cell gene expression data.

Major comments:
The recommended template is not used for the submission, so it is difficult to judge if the authors have followed the formatting requirements such as paper length and margin sizes. The main argument of the paper, that the model outputs and representational units should align with biological logics is trivial. It is not directly mentioned if the proposed method is for GRN/regulatory set recovery in a single biological context (e.g cell line) or across contexts (e.g cell types). How context similarity or dissimilarity is reflected in train/test case can substantially impact the evaluations. The evaluations as presented are therefore not confident, unless the work is re-worded to explicitly mention that the residual HOS2 model is applicable to GRN discovery in a single context only. \psi_{set}(S,t) is learned from noisy single-cell data, and may not necessarily capture the set-level interactions. That is, functional relationships between genes and the importance of context are not reflected into the proposed model.  Comparison to GENE3 as one of established GRN methods is missing.

Minor comments:
- Model architecture is stated in appendix. It is better to be moved to the main text.
- Some details such as the “label” in the multi-label objective of model training is not explained in the main text.
- A brief explanation on SERGIO DS3 (single-cell vs bulk RNA, a cell line or multiple cell types) is missing.
- Comparison to established GRN  and edge-level (comparable to results from stage 1 of the model) methods such as GENE3.
- Analysis on mESC ChIP-seq dataset should be moved to main text or at least a summary of the findings should be provided in main text.
- Can the authors show that the sets they have recovered the genes in the set exhibit high functional similarities using gene embeddings from scGPT, protein-language models such as ESM, or other experimental datasets?
- \psi_{set}(S,t) is learned from noisy single-cell data, and is not necessarily capturing the set-level interactions. Using gene embeddings from foundation models or other knowledge-bases could help here to incorporate a more “functional” notion to the recovered sets.

I'd be happy to revise the score if GENE3 or other established baselines are added, the submission is uploaded in the recommended template for a fair assessment, the author clarify if the model extends to datasets with multiple biological contexts, and if they could show that the genes recovered in the set are functionally related.

---

### Meta-Review · Area_Chair_s2KJ · 2026-02-28

**Recommendation:** Reject
**Confidence:** 3

**Metareview:**

Unfortunately this paper does not use the official LMLR template so will be reject on those grounds.

---

### Decision · Program_Chairs · 2026-03-02

**Decision:**

Reject

**Comment:**

Please see the meta-review.